# Position: AI Governance Needs ISO-like Interoperability Protocols, Not Just Laws

**Azmine Toushik Wasi** [1 2] **Mst Rafia Islam** [1 3] **Mahfuz Ahmed Anik** [1 2] **Taki Hasan Rafi** [4] **Md Manjurul Ahsan** [1 5] **Dong-Kyu Chae*** [4]

## Abstract

As Artificial Intelligence (AI) systems become deeply integrated into critical global infrastructure, the urgency for robust governance frameworks has intensified. However, current approaches, led by jurisdiction-specific laws, policies, and voluntary frameworks such as the EU AI Act, China's algorithm governance, and the NIST AI Risk Management Framework in the U.S., create a fragmented regulatory landscape. In this position paper, we argue that ***AI governance must be built not on laws alone, but on ISO-like interoperability protocols that enable standardized, machine-readable risk communication across borders***. Drawing on the success of the GDPR, which was operationalized through standards like ISO 27001 and Privacy by Design, we propose the development of standardized AI *nutrition labels* containing unified metrics for bias, energy usage, and data provenance to facilitate cross-jurisdictional compliance. These manifests would lower barriers for small and medium enterprises (SMEs), reduce redundant regulatory efforts, and build public trust. The paper addresses concerns that standards may stifle innovation by advocating for modular, versioned protocols designed to evolve in tandem with technological change. Overall, we call for a shift from siloed legal compliance toward interoperable technical conformance, enabling a shared global language for responsible AI deployment.

[1]Computational Intelligence and Operations Laboratory (CIOL), Bangladesh [2]Dept. of Industrial and Production Engineering, Shahjalal University of Science and Technology, Sylhet, Bangladesh [3]Dept. of Law, Independent University, Dhaka, Bangladesh [4]Dept. of Computer Science, Hanyang University, Seoul, South Korea [5]Dept. of Industrial and Systems Engineering, University of Oklahoma, Norman, Oklahoma, USA. Correspondence to: Dong-Kyu Chae <dongkyu@hanyang.ac.kr>.

*Proceedings of the $43^{rd}$ International Conference on Machine Learning*, Seoul, South Korea. PMLR 306, 2026. 

## 1. Introduction

The rapid adoption of Artificial Intelligence (AI), particularly in high-stakes domains such as healthcare, critical infrastructure, and finance, has amplified both its transformative potential and its systemic risks (Lekadir et al., 2025; Jenko et al., 2025). Generative AI (GenAI) systems are now integrated into organizational workflows at scale, with over 65% of enterprises reporting adoption driven by efficiency gains and new revenue opportunities (Singla et al., 2025; Databricks, 2025). This acceleration has intensified calls for governance mechanisms capable of ensuring safety, fairness, and accountability across increasingly autonomous and opaque systems (Ribeiro et al., 2025; Ligot, 2024). Yet despite broad agreement on the need for oversight, the global regulatory response remains fragmented. AI systems spanning foundation models, enterprise decision-support tools, and consumer-facing applications are deployed across jurisdictions with sharply divergent regulatory philosophies, ranging from the EU's risk-based, ex ante controls to the more market-driven and voluntary approaches prevalent in the US, despite relying on largely similar architectures, data, and risk pathways (Zhong et al., 2025; Li & Li, 2025).

This divergence is evident across major regulatory regimes. The EU AI Act establishes a tiered framework: *Unacceptable, High, Limited, and Low Risk*, with stringent conformity assessment, documentation, and post-market monitoring requirements for high-risk systems (EU, 2024; Chamberlain, 2022). China has adopted a multi-layered approach encompassing data governance, algorithm registration, cybersecurity, and ethical review, mandating filings and security assessments for systems influencing public opinion or social order (UNESCO, 2021). By contrast, the United States lacks a comprehensive federal AI statute, relying on sectoral rules, state initiatives, and voluntary guidance such as the NIST AI Risk Management Framework, which provides lifecycle-oriented principles without binding force (NIST, 2023). ASEAN has advanced non-binding regional guidance emphasizing innovation and interoperability, but without enforceable technical requirements. Together, these regimes reflect shared governance objectives but incompatible operational implementations.

This misalignment produces what we describe as a *standardization vacuum*: the absence of a shared technical language and interoperable protocols for communicating AI risk across borders (OECD, 2019). Extraterritorial regimes such as the EU AI Act[1] require global actors to interpret complex legal obligations, while China's algorithm filing and security review processes reflect distinct national priorities. Without harmonized technical standards, an AI system may be classified as *low risk* in one jurisdiction and *high risk* in another, generating compliance uncertainty, duplicated assessments, and barriers to cross-border deployment (Faveri et al., 2025). These inconsistencies are particularly acute for foundation models repurposed across enterprise and consumer contexts, where regulatory obligations shift with downstream use. The result is not merely regulatory diversity, but structural fragmentation that functions as a non-tariff barrier, disproportionately burdening small and medium-sized enterprises (SMEs) while favoring large incumbents (Journ, 2025; Zaidan & Ibrahim, 2024).

At the same time, standardization itself carries political and economic risk. History shows that technical standards can become non-tariff barriers when captured by incumbents, tied to proprietary tooling, or associated with compliance costs that smaller actors cannot absorb (Blind, 2016; OECD, 2017; UNCTAD, 2021). In such cases, standards intended to promote safety and trust may entrench market concentration and exclude firms from the Global South or SMEs with limited regulatory capacity (Funk & Methe, 2001). To avoid this outcome, AI interoperability standards must be open, modular, and extensible, supported by transparent governance processes and low-cost implementation pathways. Capacity-building, reference implementations, and public infrastructure are therefore not peripheral concerns but core requirements for equitable global AI governance (Krechmer, 2006).

In this paper, we argue that ***AI governance requires ISO-like technical interoperability protocols, not legal frameworks alone, to achieve global accountability, trust, and responsible innovation.*** Laws are indispensable for setting normative thresholds and enforcement authority, but they are insufficient to ensure consistent operationalization across heterogeneous AI systems and jurisdictions. Rather than replacing existing regulatory regimes, we argue for a complementary technical layer that enables shared, machine-readable representations of AI risk. We propose standardized AI nutrition labels as modular, versioned manifests encoding comparable metrics for bias, energy consumption, and data provenance, enabling AI systems to carry a form of interoperable compliance credential across borders. Drawing on precedents such as ISO 27001 (ISO/IEC, 2023a) and international product safety certification, we articulate a governance model in which laws define *what* constitutes acceptable AI behavior, while technical standards specify *how* compliance is demonstrated in practice. This approach is explicitly designed to support domain-specific variation while preserving stable interoperability primitives, providing a scalable foundation for global AI governance.

## 2. Background: GDPR's Success Through Standardized Implementation

The General Data Protection Regulation (GDPR)[2], enacted by the European Union, fundamentally reshaped the global data protection landscape by establishing harmonized privacy obligations with extraterritorial reach (Gebru et al., 2021; Cohen, 2019). Its primary objective was to strengthen individual control over personal data while creating a unified regulatory framework across EU member states (Terlecki, 2023). The regulation's global impact was reinforced by substantial penalties for non-compliance, a feature echoed by the EU AI Act's proposed fines of up to €35 million or 7% of global turnover (NAVEX). As a result, organizations worldwide were compelled not only to understand GDPR's legal requirements, but also to operationalize them within complex technical and organizational environments. GDPR thus provides a salient example of how ambitious regulation can drive global governance change when supported by effective implementation mechanisms.

**Operationalization through ISO 27001 and Privacy by Design.** While GDPR articulated legal obligations for lawful, fair, and secure data processing, ISO 27001[3] supplied a practical and internationally recognized framework for implementing these requirements through an Information Security Management System (ISMS) (ISO/IEC, 2023b). Alignment with ISO 27001 has been widely recognized as facilitating GDPR compliance by structuring risk assessment, security controls, documentation, and accountability processes. In parallel, the principle of *Privacy by Design* (PbD), introduced by Ann Cavoukian in 1995 and codified in GDPR Article 25, mandated the proactive integration of privacy safeguards into system architectures and organizational workflows (Wilkinson et al., 2016; Cavoukian, 2009). Together, ISO 27001 and PbD transformed GDPR from an abstract legal mandate into a set of concrete, auditable, and repeatable operational practices spanning the full data lifecycle.

**Limits of the GDPR analogy for AI governance.** Despite its success, the GDPR–ISO 27001 model cannot be transferred wholesale to AI governance. Unlike personal data flows, AI systems are opaque, adaptive, and probabilistic, with behavior that changes over time and across

[1] https://artificialintelligenceact.eu

[2] https://gdpr-info.eu/

[3] https://www.iso.org/standard/27001

*Table 1.* GDPR-ISO 27001/Privacy by Design Overlap and Benefits

| GDPR Requirement/Principle | Corresponding ISO 27001 / Privacy by Design Principle | Operational Implication / Benefit |
| --- | --- | --- |
| Data Protection by Design & Default | Proactive Not Reactive, End-to-End Security | Privacy embedded from initial design, continuous protection across data lifecycle. |
| Data Minimization | A.14 System acquisition/development/maintenance | Reduced data exposure, processing only necessary personal data. |
| Lawfulness, Fairness, Transparency | A.5 Information Security Policies, Clear Documentation | Clear data handling processes, increased visibility and trust. |
| Accountability | Systematic Risk Assessment, Defining Roles & Responsibilities | Documented procedures, clear ownership for data protection. |
| Security of Processing | A.9 Access Control, Data Encryption, Incident Response Protocols | Robust technical and organizational safeguards, effective incident management. |
| Data Subject Rights | Operational Procedures for Data Subject Rights | Streamlined processes for fulfilling individual privacy requests. |
| **Overall Impact** | Unified Information Security Management System: Enhanced reputation and advantage, process optimization, increased efficiency. | |

deployment contexts (Mittelstadt, 2019). While data protection primarily governs access, processing, and storage of identifiable information, AI governance must also address model behavior, downstream adaptation, emergent risks, and domain-specific performance variability that cannot be fully specified ex ante (Chamberlain, 2022). Critically, ISO 27001 governs organizational processes, how a company manages information security, whereas AI manifests must govern product outputs: how a deployed model behaves. Unlike traditional IT security processes, AI behavior is stochastic and emergent, meaning that the same model may produce different outputs under distributional shift or novel prompts not covered at certification time. Consequently, whereas ISO 27001 operationalizes relatively stable security objectives (Terlecki, 2023), AI governance standards must contend with greater technical uncertainty, faster iteration cycles, and heterogeneous deployment environments (Perboli et al., 2025). GDPR-style compliance mechanisms are therefore instructive but insufficient for AI without complementary and evolvable technical standards (Parnas, 1972).

**Governance lesson: coupling law with technical standards.** GDPR's effectiveness stemmed not from legal authority alone, but from reinforcement through interoperable technical standards and design principles (ISO/IEC, 2023b; OpenStand Principles, 2012). This experience highlights a broader lesson: legal frameworks define the *what* of compliance, while technical standards operationalize the *how* (NIST, 2023; Moreau et al., 2013). In the AI context, this lesson applies with caution rather than equivalence: laws may establish risk thresholds and accountability obligations (EU, 2024; Chamberlain, 2022), but interoperable technical standards are required to express these requirements as machine-readable, auditable artifacts across jurisdictions and AI domains (Cintron et al., 2024; Faveri et al., 2025). This directly motivates our central position that AI governance must extend beyond law alone toward ISO-like interoperability protocols for risk communication and verification (Omdia, 2021; Blind, 2016). A closely parallel precedent is the Software Bill of Materials (SBOM) (Siddiqui, 2025), a machine-readable inventory of software components that became a de facto supply-chain security requirement through US Executive Order 14028 and subsequent procurement mandates rather than through treaty negotiation. Like the AI Risk Manifest we propose, SBOMs are standardized, versioned artifacts that travel with a product and enable downstream auditing without requiring the auditor to inspect internal source code. This supply-chain adoption model, in which a technical artifact becomes mandatory through procurement rather than law, is the precise diffusion mechanism we advocate for AI governance.

Table 1 summarizes how GDPR requirements were operationalized through ISO 27001 controls and Privacy by Design principles, illustrating the complementary relationship between legal mandates and technical standards. The table highlights how abstract regulatory obligations were translated into concrete processes, audits, and safeguards, reinforcing the role of standards as essential instruments for scalable compliance rather than optional best practices. This precedent supports our argument that effective AI governance similarly requires a coupling of legal authority with interoperable, operational technical standards.

## 3. Machine-Readable AI Risk Manifests

### 3.1. Concept of AI *Nutrition Labels*

The concept of AI *nutrition labels* has gained traction as a means of simplifying the communication of complex AI risks, analogous to how food labels summarize ingredients and nutritional content (Gerke, 2023). Across industry and policy discussions, such labels are promoted to enhance transparency, build user confidence, and support informed decision-making about AI capabilities and limitations. In this work, however, we use the term *AI nutrition label* not merely as a metaphor, but to denote a **structured, machine-readable AI risk manifest** that encodes standardized information about a system's properties, limitations, and compliance-relevant characteristics. This reframing from descriptive summaries to technical artifacts is essential for interoperability across organizational, technical, and regulatory boundaries.

Early initiatives illustrate this emerging practice. Omnissa's AI labels disclose model type, provider, data sources, data sovereignty, and training data usage (Smith, 2025), while CHAI's Applied Model Cards are increasingly used in

healthcare to present baseline information about deployed AI tools (Mitchell et al., 2019). These efforts streamline procurement and high-level regulatory review by condensing extensive documentation into accessible summaries. However, most existing labels remain primarily human-readable artifacts, lacking standardized schemas, explicit versioning, and formal mappings to regulatory or risk management frameworks. We therefore advance a more formal interpretation of AI nutrition labels as technically specified manifests, analogous to software bills of materials, that can be parsed, validated, and compared programmatically, enabling integration into MLOps pipelines, procurement systems, and regulatory workflows while preserving the communicative clarity of the nutrition label metaphor.

### 3.2. Unified Metrics for Key AI Risk Dimensions

Interoperable AI governance requires a *minimum viable set of standardized metrics* that can be reported, compared, and verified across jurisdictions and deployment contexts. Rather than enforcing a single definition of risk, such metrics provide a shared technical vocabulary through which heterogeneous regulatory requirements can be operationalized. We focus on three recurring dimensions: bias, energy consumption, and data provenance, which appear across legal, ethical, and policy frameworks. To enable machine-readability and automated compliance, these dimensions must be expressed through structured schemas with explicit reporting requirements. Accordingly, we adopt ISO-style *must* and *should* language to emphasize implementability.

**Bias measurement and reporting (fairness).** Algorithmic bias, often originating from non-representative data or subjective annotation, can produce discriminatory outcomes in domains such as hiring, lending, and law enforcement (Bolukbasi et al., 2016). To support consistent evaluation, AI systems *must* report at least one global fairness metric and one subgroup-based metric, such as Equalized Odds or Disparate Impact, using a standardized schema (Feldman et al., 2015; Hardt et al., 2016; Speicher et al., 2018). Existing toolkits, including IBM AI Fairness 360, Google's What-If Tool, and Microsoft Fairlearn, demonstrate the feasibility of such reporting (Varshney, 2018; Wexler et al., 2019; Bird et al., 2020). Given the absence of a universally correct fairness definition and frequent constraints on protected attributes, manifests *should* document metric selection, proxy use, and known limitations (IEEE, 2021). Formal initiatives such as the Indian Telecommunication Engineering Centre's fairness assessment framework further illustrate how thresholds and scenario testing can be incorporated into standardized reporting (TEC, 2023).

**Energy consumption and environmental impact.** The environmental footprint of AI systems, particularly large-scale models, has emerged as a major governance concern due to energy use, carbon emissions, and water consumption (Strubell et al., 2019). AI systems *must* report inference-time energy consumption under a standardized evaluation setting that specifies task, hardware, and measurement protocol, enabling meaningful cross-system comparison (Anthony et al., 2020). Efforts such as the AI Energy Score demonstrate how energy efficiency can be communicated through comparable metrics and ratings (Wu et al., 2022). Where feasible, manifests *should* also disclose training-related emissions and carbon intensity to support sustainability-aware procurement and oversight (Schwartz et al., 2020). Standardized energy reporting enables environmental criteria to be integrated directly into regulatory review and public-sector acquisition. For distilled models, the manifest must include a `teacher_model_ref` field within the `energy_use` block, linking to the teacher model's manifest and its associated training energy. The distilled model then reports its own (substantially lower) inference costs separately. This preserves full traceability of the AI supply chain while correctly attributing the amortized environmental cost of knowledge distillation.
Regarding reporting scope, the schema mandates at minimum the energy cost of the final training run of the deployed checkpoint. Optional structured fields allow disclosure of cumulative energy across experimental checkpoints, hyperparameter search runs, and ablations, enabling auditors to assess total lifecycle cost where that information is available.

**Data provenance and traceability.** Data provenance, defined as the documented lineage of data origins, transformations, and usage, is essential for accountability and trust in AI systems, particularly large language models whose behavior depends heavily on training data quality (Moreau et al., 2013; Crilly, 2025). AI systems *must* provide a dataset lineage summary describing data sources, geographic scope, and applicable licensing or usage constraints. Traceability mechanisms based on established standards such as W3C PROV and ISO provenance practices *should* support auditing of how data and model outputs are generated and modified over time (Moreau et al., 2013; Osarenren, 2024). Techniques including watermarking, telemetry, blockchain records, and metadata management systems can operationalize these requirements. Provenance reporting is critical for attributing responsibility, identifying harmful outputs, and managing copyright or bias risks across the AI supply chain. Security-oriented extensions to the provenance block can include `data_filtering_pipeline`, `poisoning_detection_methods`, and `dataset_integrity_checks` fields, documenting both preventive controls and empirical validation results such as adversarial contamination test pass rates or certified dataset integrity scores.

Collectively, these unified metrics do not harmonize legal thresholds but establish a shared technical substrate for AI risk communication. By mandating comparable reporting of fairness, energy use, and data lineage, machine-readable manifests allow jurisdiction-specific obligations to be interpreted through a common interoperability layer. This specification-oriented approach reinforces our central position that scalable AI governance depends not only on legal mandates, but on standardized, auditable metrics that translate abstract risk principles into operational practice.

### 3.3. Machine-Readable Schemas for AI Tools

The fragmentation of AI governance across the EU, United States, China, and ASEAN highlights the absence of a shared technical language for expressing AI risk and compliance (Perboli et al., 2025). Regulatory approaches span the EU's binding, risk-tiered AI Act, China's multi-layered governance framework (Midfa, 2025), ASEAN's non-binding generative AI principles (ASEAN, 2025), and the United States' voluntary NIST AI Risk Management Framework[4]. For organizations deploying AI systems across borders, this diversity produces duplicated assessments, inconsistent documentation, and high compliance overhead. Addressing these challenges requires machine-readable schemas that support automated exchange, validation, and comparison of governance information across regulatory contexts. Formats such as JSON and YAML are well suited to this role, combining API-level interoperability with human-readable structure for complex configurations.

**Existing documentation and transparency initiatives.** A range of initiatives has emerged to improve AI transparency, including IBM AI FactSheets for lifecycle metadata tracking, Google's Model Card Toolkit for structured model documentation, and OpenAI's Model Spec for defining intended behavior and deployment constraints. In the public sector, the UK's Algorithmic Transparency Recording Standard enables standardized disclosure of algorithmic tools and risks (UK Cabinet Office, 2023), while the OECD's voluntary reporting framework and IEEE CertifAIEd address responsible AI practices and ethical assessment. Recent work such as the ATLAS framework (Spoczynski et al., 2025) demonstrates how ML lifecycle provenance and transparency can be operationalized at the artifact level, offering a concrete technical reference point for the manifest design we propose. Despite this progress, these efforts remain fragmented and largely focused on human-readable or organizational reporting. None integrates multiple risk dimensions, such as fairness, energy consumption, data provenance, and regulatory alignment, into a single, versioned, machine-readable artifact, leaving compliance evidence siloed and poorly suited for automated verification or cross-jurisdictional reuse.

[4]https://www.nist.gov/itl/ai-risk-management-framework

**Toward interoperable AI risk schemas.** Table 2 illustrates how divergent regional governance regimes impose incompatible risk classifications and compliance obligations, making the *standardization vacuum* tangible. The comparison highlights that fragmentation is not merely legal, but technical: there is no shared schema through which risk information can be consistently expressed and interpreted. A machine-readable AI Risk Manifest addresses this gap by standardizing the representation of core governance fields, such as risk classification, intended use, audit artifacts, monitoring metrics, and jurisdictional references, without harmonizing legal thresholds. The manifest layer does not override jurisdiction-specific requirements; rather, it provides a shared technical substrate through which heterogeneous obligations can be operationalized.

In the absence of such a standardized schema, governance information itself becomes opaque, limiting cross-border accountability, automated compliance checks, and meaningful comparison. We therefore argue that an ISO-like, machine-readable schema designed explicitly for interoperability, rather than organizational reporting alone, is necessary to transform existing documentation efforts into a shared technical infrastructure for global AI governance. This shift reframes AI transparency from fragmented disclosure toward reusable, verifiable compliance objects capable of functioning across jurisdictions and AI domains.

### 3.4. Proposed ISO-like Schema

We propose a globally recognized, ISO-like schema for AI risk manifests that provides a unified framework for structured, verifiable, and machine-readable risk communication. The objective is not to replace existing legal regimes, but to enable their consistent operationalization across jurisdictions through a shared technical specification. The schema defines a standard set of documentation fields that reflect common accountability goals while remaining modular and extensible to domain-specific requirements. By formalizing these fields, AI risk information can be expressed, exchanged, and validated consistently across organizational, technical, and regulatory boundaries.

**Core schema components.** At a minimum, an AI risk manifest should include the following components. First, *model identification* captures a unique system identifier, versioning information, and accountable developer and deployer entities, ensuring traceability across the AI lifecycle. Second, *purpose and deployment context* specifies intended use, out-of-scope applications, user populations, and domain constraints, helping to prevent misuse or inappropriate deployment. Third, *data provenance* documents training and evaluation data sources, geographic scope, collection methods, and preprocessing steps, enabling legal, ethical, and bias-related assessment of data use (Calmon et al., 2017).

*Table 2.* Comparison of major AI governance regimes and how a machine-readable AI Risk Manifest standardizes risk communication.

| Jurisdiction | Regulatory Approach | Risk / Compliance Focus | What a Manifest Would Standardize |
|---|---|---|---|
| EU | Binding, risk-based regulation (EU AI Act) | Tiered risk classification; conformity assessment; post-market monitoring | `risk_classification`, `intended_use`, `audit_artifacts`, `monitoring_metrics` |
| China | Algorithm registration and security governance | Algorithm filing; data security; public opinion and social risk | `system_id`, `deployment_context`, `jurisdictional_registry_refs`, `security_controls` |
| United States | Sectoral, voluntary, lifecycle-oriented guidance | Risk management practices across development and deployment | `risk_management_profile`, `evaluation_metrics`, `governance_controls` |
| ASEAN | Non-binding regional guidance | Harmonization, innovation enablement, trust-building | `baseline_risk_summary`, `transparency_fields`, `versioned_schema` |

Fourth, *performance and limitations* report accuracy, robustness, uncertainty, and known failure modes, ensuring that system behavior is communicated transparently and responsibly.

Beyond these baseline fields, the schema incorporates cross-cutting governance dimensions. *Bias and fairness* reporting encodes standardized fairness metrics and mitigation strategies, supporting comparative evaluation across demographic groups. *Energy and resource consumption* captures training and inference energy use, carbon footprint estimates, and efficiency optimizations, embedding sustainability into AI evaluation. To mitigate hardware-induced comparability bias, manifests must disclose the full measurement context: hardware type and configuration, utilization rates, workload specification, and measurement methodology. This context-aware disclosure allows evaluators to normalize or reinterpret energy metrics across hardware families rather than relying on a single aggregate figure that may disadvantage architectures optimized for non-standard hardware. *Security and safety* fields document known vulnerabilities, threat models, and mitigation strategies, while *transparency and explainability* record the availability of interpretability tools and human oversight mechanisms. Finally, *ethical considerations* summarize misuse risks, dual-use concerns, and internal governance processes, framing ethics as an ongoing responsibility rather than a static list.

**Regulatory alignment and interoperability.** A defining feature of the proposed schema is an explicit *regulatory alignment* section that maps the system's risk profile to applicable governance frameworks, such as the EU AI Act risk tiers or NIST AI RMF functions. This crosswalk enables a single manifest to function as a reusable compliance artifact across jurisdictions, reducing redundant documentation and easing cross-border deployment. Importantly, the schema does not harmonize legal thresholds; instead, it standardizes how risk evidence is represented and communicated, allowing regulators to apply their own enforcement logic to a common technical substrate.

The shift from human-readable documentation to machine-readable schemas transforms compliance from a manual, interpretative process into an auditable and automatable workflow (Gebru et al., 2021). Standardized manifests can be ingested by regulatory or organizational systems for validation, monitoring, and post-market auditing, increasing efficiency while reducing compliance costs (Raji et al., 2020). In this sense, AI risk manifests function as digital containers for governance information, analogous to how standardized shipping containers enabled scalable global trade. By lowering barriers to entry and reuse, interoperable manifests allow developers to focus on building robust AI systems while enabling regulators, procurers, and users to make informed and trustworthy decisions.

**Illustrative example.** To ground the schema in practice, Listing 1 presents a compact, illustrative example of a machine-readable AI nutrition label in JSON format.

*Listing 1.* Illustrative AI nutrition label (short-form, JSON).

```
{
  "model_id": "RawModel-1",
  "purpose": "Healthcare triage decision
      support",
  "data_provenance": {
    "source": "clinical notes",
    "region": "EU hospitals"
  },
  "bias_metrics": {
    "equalized_odds_score": 0.92,
    "disparate_impact_ratio": 0.87
  },
  "energy_use": {
    "training_co2_tons": 75,
    "inference_kwh_per_1k": 0.05
  },
  "limitations": [
    "not validated for pediatric patients"
  ],
  "regulatory_alignment": {
    "eu_ai_act_risk_tier": "high_risk",
    "nist_ai_rmf_function": "MANAGE"
  }
}
```

A full protocol-level worked example of a machine-readable

AI Risk Manifest, including JSON and YAML representations, cryptographic attestation hooks, and a regulatory crosswalk, is provided in Appendix A.

**Threat model and verifiability.** A central risk of any documentation-based governance mechanism is self-reporting bias, in which model developers selectively disclose favorable metrics or evaluate on cherry-picked test sets, leading to "compliance theater" rather than substantive accountability. It is important to distinguish two properties: *integrity* and *truthfulness*. Cryptographic attestations (e.g., signed hashes) guarantee integrity, that reported values have not been altered after signing, but they cannot by themselves guarantee truthfulness, i.e., that the underlying evaluation was conducted on a representative and unmanipulated test set. The manifest is therefore designed as a minimum verifiable baseline rather than a truth oracle. To mitigate the truthfulness gap, key fields (`data_provenance`, `evaluation`, and the attestation block) are designed to be cross-verified by independent auditors or certified testing bodies who can re-run evaluations against the same hashed dataset artifacts without re-training the model. Under this design, regulators and downstream deployers are not required to trust the model producer's claims directly, but only the integrity of the verification chain. This shifts AI governance from narrative transparency to evidence-backed, machine-checkable assurance. A detailed treatment of verification architecture, including incremental assurance layers and third-party attestation protocols, is provided in Appendix C.

**Mitigating Goodhart's Law (Goodhart, 1984).** A related risk is that developers optimize their models to pass manifest checks without genuinely reducing risk, a form of adversarial compliance. The manifest design addresses this through three mechanisms. First, rather than mandating a single scoring rule, the schema requires disclosure of the specific metric selected and the rationale for its selection, making metric-gaming visible to auditors. Second, following ISO audit practice, compliance is subject to continuous review and certificate revocation when post-market monitoring data (e.g., `drift_auc` or `incident_rate`) diverges from certified values. Third, the schema supports dynamic auditing via API-accessible telemetry, enabling probabilistic third-party spot checks on live model behavior rather than relying solely on static certification data.

## 4. Policy Framework: Linking Regulatory Compliance to Open Standards

The global AI regulatory landscape is highly fragmented, with major economies adopting divergent governance frameworks (Chun et al., 2024). The EU AI Act imposes binding obligations on high-risk AI systems, including conformity assessments and technical documentation, while China mandates algorithm registration and security reviews for systems influencing public opinion (Creemers et al., 2022). In contrast, the United States relies primarily on voluntary guidance such as the NIST AI Risk Management Framework, and ASEAN promotes non-binding principles aimed at regional harmonization. These divergent approaches generate overlapping and often inconsistent compliance requirements, increasing costs and slowing cross-border AI deployment. We argue that open, collaboratively developed technical standards provide a practical mechanism for linking these heterogeneous regulatory regimes without harmonizing their legal thresholds.

**Regulation reinforced by open standards.** Precedents from other regulated domains demonstrate that regulation and open standards can be mutually reinforcing. In Open Banking, PSD2 and the UK Open Banking Standard mandate interoperable APIs, enabling innovation while preserving regulatory oversight (EU, 2015). Energy systems similarly rely on NIST-led Smart Grid interoperability standards to ensure secure coordination across heterogeneous infrastructure (NIST, 2009). IoT governance increasingly incorporates standardized Software Bills of Materials to support transparency and security (Siddiqui, 2025), while digital identity ecosystems depend on open standards such as W3C Verifiable Credentials and FIDO2 (Giannopoulou, 2023). Cybersecurity frameworks like the NIST CSF have also become de facto regulatory benchmarks, illustrating how technical standards can operationalize high-level policy goals (Siddiqui, 2025). Together, these examples suggest a scalable model for AI governance in which legal mandates are translated into interoperable technical requirements.

**Mechanisms for integrating standards into AI regulation.** Several complementary mechanisms can link adherence to open technical standards with regulatory compliance. For high-risk systems, regulators may mandate conformance with ISO-like standards, such as ISO/IEC 42001 for AI management systems, analogous to CE marking in product safety regimes, though experience suggests that poorly calibrated mandates can generate unintended strategic behavior rather than substantive safety gains (Laufer et al., 2025). For lower-risk or rapidly evolving domains, a *comply or explain* approach offers greater flexibility by allowing deviations from standards provided they are transparently justified, aligning regulatory interpretation with system-specific context and evolving technical realities (He et al., 2025). Regulatory sandboxes can further support iterative testing of emerging standards and safety practices, including structured red-teaming and evaluation protocols, though care is required to ensure that such environments do not disproportionately advantage well-resourced actors (Deng et al., 2025). Finally, embedding open standards into public-sector procurement can leverage buyer power to shape market norms and encourage safer system behavior by design, particularly when coupled with mechanisms that

enable systems to decline unsafe actions or exit hazardous operational states (Bonagiri et al., 2025).

Taken together, these mechanisms enable incremental adoption of interoperable standards while acknowledging political, economic, and institutional constraints. They do not eliminate global regulatory fragmentation, but provide pragmatic pathways for convergence around shared technical representations of AI risk. In this framework, open standards function as a coordination infrastructure rather than as substitutes for law, supporting innovation while strengthening accountability (Manski, 2022).

### 4.1. Bridging Technical Standards with Policy

Legal and ethical frameworks provide essential guardrails for AI governance, but they are insufficient for globally deployed and rapidly evolving systems, and may even backfire when regulatory signals are weak or incomplete (Laufer et al., 2025). Effective oversight therefore requires interoperable technical standards that translate statutory intent and policy goals into operational, verifiable practice (He et al., 2025). The proposed AI risk manifest schema addresses this gap by offering a standardized, machine-readable artifact that functions as both technical documentation and a reusable compliance instrument. When embedded in certification regimes, regulatory sandboxes, and procurement processes, such manifests shift governance from abstract principles toward implementation-focused accountability, complementing emerging technical approaches to AI safety testing, privacy protection, and agent-level control (Deng et al., 2025; Ashiq et al., 2025; Bonagiri et al., 2025).

Globally harmonized technical standards for AI risk communication offer concrete benefits, including enhanced accountability through traceable documentation (Crilly, 2025), reduced friction in cross-border deployment (Hamzah, 2025), safer and faster innovation through predictable design constraints (Martin, 2025), and increased public trust via explainable and accessible reporting (Smith, 2025). These benefits illustrate how technical interoperability complements, rather than replaces, legal oversight. We therefore contend that sustainable and equitable AI governance must be grounded not only in law, but in an infrastructure of trust built on standardized, interoperable, and verifiable technical foundations, much as ISO protocols underpin safety in traditional engineering domains (Omdia, 2021).

## 5. Discussion

The jurisdiction-specific design of existing frameworks produces a persistent *standardization vacuum* that undermines interoperability, raises compliance costs, and complicates cross-border accountability (Faveri et al., 2025). Drawing on the GDPR experience, we show that legal authority alone is insufficient for scalable AI governance. Effective oversight requires interoperable technical standards that translate regulatory intent into implementable and auditable practice (Mittelstadt, 2019).

Our central contribution is the proposal of machine-readable, modular, and versioned AI risk manifests that function as standardized *nutrition labels* for AI systems. By integrating dimensions such as fairness, energy use, data provenance, and regulatory alignment into a reusable artifact, these manifests support consistent risk communication across jurisdictions while reducing redundant compliance costs.

A major challenge is semantic fragmentation: jurisdictions may adopt incompatible fairness definitions, such as equal opportunity or demographic parity. Rather than resolving these conflicts, the manifest standardizes their *representation* by requiring explicit reporting of metric definitions, subgroup specifications, measured values, and jurisdictional thresholds. The same JSON artifact can thus be interpreted under different legal regimes, functioning as a neutral, machine-readable reporting layer.

We acknowledge that global AI standardization is constrained by political competition, uneven institutional capacity, and rapid technological change, and we therefore emphasize governance through modularity, versioning, and iterative revision under ISO-led, multi-stakeholder stewardship involving organizations such as IEEE, NIST, and the OECD, with transparent governance and Global South participation remaining essential for legitimacy. Finally, effective AI governance depends not only on interoperable standards but also on verifiability: treating risk manifests as cryptographically attestable compliance objects enables accountability without blind trust in self-reporting, while verification chains, third-party attestations, and immutable audit references raise the cost of misrepresentation. Together, these mechanisms support a shift from reactive, jurisdiction-specific compliance toward governance by design, embedding safety, fairness, and accountability directly into AI development and deployment.

## 6. Alternative Views

Here, we explore two common objections to interoperability-based AI governance:

**Alternative View 1: Standards Inevitably Lag Behind Innovation.** A common argument is that technical standards will slow AI innovation (details in Appendix §D) because they cannot keep pace with the rapid evolution of models, data, and deployment practices (Cohen, 2019). As AI systems increasingly update through continuous learning and deployment, critics contend that any fixed standard risks becoming obsolete before it is widely adopted (Harvard Law Today, 2025). This concern is amplified by fears that early or rigid standardization could lock in suboptimal design choices and discourage experimentation. From this perspective, innovation is best preserved by minimizing for-

mal constraints and allowing practices to evolve organically. However, this view assumes that the absence of standards preserves agility, overlooking the coordination costs imposed by fragmented requirements at scale.

**Alternative View 2: Standards Risk Entrenching Incumbents and Raising Entry Barriers.** A related critique holds that global standards disproportionately benefit large technology firms with established compliance capacity, while imposing burdens that smaller firms and new entrants struggle to absorb (Heller, 2023). Compliance with formal schemas, audits, and documentation processes may require legal, technical, and financial resources unavailable to startups or researchers. Critics warn that this dynamic could strengthen incumbents, reduce competition, and slow the diffusion of innovation, particularly in fast-moving AI markets (Ananny & Crawford, 2018). In this view, standards function less as neutral infrastructure and more as gatekeeping mechanisms. Yet this argument underestimates how uncoordinated regulatory fragmentation already imposes higher relative costs on smaller actors forced to navigate multiple incompatible regimes.

**Implication for our position.** The core issue is not the existence of standards but their design. Modular, versioned, outcome-oriented standards can evolve with AI systems while preserving technical freedom by specifying *what* must be demonstrated rather than *how*. By reducing redundant compliance and coordination costs, interoperable standards act as enabling infrastructure rather than constraints, and ISO-like protocols become prerequisites for scalable responsible AI across borders. We acknowledge a tradeoff: overly frequent updates can create compliance fatigue and certification churn. This can be managed through a multi-stakeholder body (e.g., ISO, IEC, and IEEE) that releases major schema versions on multi-year cycles while allowing localized modular extensions for faster domain-specific updates, similar to how ISO standards maintain stable cores with evolving annexes.

## 7. Recommendations and Call to Action

To move from fragmented compliance toward interoperable AI governance, we outline concrete actions for policymakers, standards bodies, and the research community:

**R1: Establish a shared technical baseline for AI risk communication.** Policymakers and standards bodies should prioritize the development of a minimal, machine-readable baseline for AI risk manifests that can be reused across regulatory regimes. Anchoring this effort in existing institutions such as ISO and IEC would avoid governance duplication while enabling jurisdiction-specific enforcement (Omdia, 2021). Without a shared baseline, fragmentation will continue to incentivize regulatory arbitrage.

**R2: Tie interoperability standards to incentives, not only mandates.** Governments, funding agencies, and large procurers should reward adherence to interoperable AI risk standards through procurement criteria, certification pathways, and research evaluation norms. Precedents from cybersecurity and supply-chain governance show that incentive-based adoption can accelerate standard uptake while preserving flexibility (Manski, 2022). This approach lowers entry barriers for smaller actors by replacing multiple bespoke requirements with a single reusable artifact.

**R3: Invest in inclusive, multi-stakeholder standard stewardship.** To prevent standards from becoming exclusionary or incumbent-driven, governance processes must include formal mechanisms for transparency, balanced participation, and capacity building, particularly for low-resource contexts (Cihon et al., 2020). Academic institutions, civil society, and open-source communities should play a sustained role in reviewing and evolving technical specifications. Without such safeguards, standards risk reproducing the very inequities they are intended to mitigate.

**R4: Fund and formalize the science of AI evaluation.** The utility of an interoperable AI Risk Manifest depends on the scientific validity and reproducibility of the metrics it encodes. Unlike traditional software, AI systems are opaque, adaptive, and probabilistic, making static evaluation difficult. Current methods for assessing emergent behaviors such as prompt injection, socio-technical bias, and agentic drift remain fragmented and underdeveloped. Policymakers, funding agencies, and industry consortia should invest in the foundational science of AI evaluation, including robust mathematical benchmarks for fairness, hardware-agnostic protocols for energy measurement, and rigorous automated red-teaming methods. Without scientifically grounded and hard-to-game metrics, the manifest risks enabling compliance theater rather than meaningful accountability. Therefore, schema standardization must be coupled with sustained investment in the evaluation science that informs it.

## 8. Conclusion

As artificial intelligence systems become globally deployed, continuously updated, and increasingly embedded in high-stakes domains, governance mechanisms grounded in law alone cannot scale. We argue that **effective AI governance requires ISO-like interoperability protocols that complement law by enabling standardized, verifiable, and cross-border risk communication.** We propose to operationalize this through machine-readable AI risk manifests (*nutrition labels*), which are necessary to translate regulatory intent into verifiable, cross-border practice. By standardizing risk information expression rather than imposing uniform legal thresholds, this approach preserves national autonomy while reducing regulatory fragmentation. We argue that interoperable technical standards should be treated as core governance infrastructure, not documentation, enabling accountability and trust to scale across jurisdictions.

## Acknowledgements

This work was supported by the Institute of Information & communications Technology Planning & Evaluation(IITP) grant funded by the Korea government(MSIT)(**RS-2025-25422680**, Metacognitive AGI Framework and its Applications, and **RS-2020-II201373**, Artificial Intelligence Graduate School Program (Hanyang University)).

## Conflict of Interest Disclosure

The authors declare that they have no financial or other substantive conflicts of interest that could reasonably be perceived to influence this work.

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

## A. Detailed Worked Example of AI Nutrition Label

A key advantage of ISO-like interoperability protocols is that they transform compliance evidence from narrative documents into machine-readable, versioned objects that can be exchanged across borders and automatically validated. Concretely, we propose an AI Risk Manifest (a nutrition label) as a signed, structured payload that communicates: (i) system identity and intended use, (ii) risk tiering and applicable regimes, (iii) standardized metrics (fairness, performance, energy, privacy/security), and (iv) operational controls (monitoring, incident reporting, audit hooks). Below we provide a worked example of a minimal manifest for a high-impact decision support system (illustrative: automated résumé screening for hiring support). The example is intentionally compact (v1) while preserving extensibility through semantic versioning and optional extension blocks.

*Listing 2.* Worked example of an AI Nutrition Label / Risk Manifest (JSON).

```
{
  "schema": "nutrition-label.ai/manifest",
  "schema_version": "1.0.0",
  "manifest_id": "nutrition-label:demo:v1",
  "issued_at": "2026-01-15T00:00:00Z",
  "issuer": {
    "org": "ExampleAI Ltd.",
    "contact": "compliance@example.ai"
  },

  "system": {
    "name": "RawModel-1",
    "system_version": "2.3.1",
    "model_type": "LLM+ranker",
    "model_fingerprint": "sha256:7c3b...e91a",
    "deployment": {
      "mode": "API",
      "regions": ["EU", "US", "BD"]
    }
  },

  "intended_use": {
    "purpose": "Decision support for recruiter triage of resumes",
    "users": ["HR analysts"],
    "decision_impact": "employment",
    "human_in_the_loop": true,
    "prohibited_use": [
      "fully-automated hiring decisions without human review",
      "use outside declared job families without re-validation"
    ]
  },

  "risk_classification": {
    "eu_ai_act_style": {
      "tier": "high_risk",
      "rationale": "employment decision support"
    },
    "nist_rmf_profile": {
      "functions": ["GOVERN", "MAP", "MEASURE", "MANAGE"]
    }
  },

  "data_provenance": {
    "training_sources": [
      {
        "type": "public_job_postings",
        "region_scope": ["EU", "US"],
        "license": "mixed"
      },
      {
        "type": "historical_resumes",
```

```
      "region_scope": ["EU"],
      "license": "internal_consent_basis"
    }
  ],
  "sensitive_attributes_used": false,
  "known_gaps": [
  "under-representation of candidates with non-traditional education"
  ],
  "security": {
      "data_filtering_pipeline": "deduplication + PII scrub + toxicity filter v2.1",
      "poisoning_detection_methods": ["activation_clustering", "spectral_signatures"],
      "dataset_integrity_checks": "SHA-256 per shard, verified at load time"
  }
},

"evaluation": {
  "eval_date": "2026-01-10",
  "datasets": [
    {
      "name": "HR-Triage-2025Q4",
      "region": "EU",
      "n": 50000,
      "split": "heldout"
    }
  ],
  "metrics": {
    "utility": {
      "topk_precision@10": 0.61,
      "auc": 0.79
    },
    "robustness": {
      "ood_drop_auc": 0.06,
      "prompt_injection_pass_rate": 0.93
    }
  }
},

"fairness": {
  "reporting_unit": "EU legal categories where available",
  "group_metrics": [
    {
      "group": "gender",
      "metric": "equal_opportunity_diff",
      "value": 0.04,
      "threshold": 0.05
    },
    {
      "group": "age_bucket",
      "metric": "selection_rate_ratio_min",
      "value": 0.82,
      "threshold": 0.80
    }
  ],
  "mitigations": [
    "post-hoc calibration",
    "bias-aware re-ranking"
  ]
},

  "energy": {
      "training_kwh": 42000,
      "inference_wh_per_1k_tokens": 2.1,
      "measurement_protocol": "nutrition-label.ai/energy/v1",
      "measurement_context": {
         "hardware": "NVIDIA A100-80GB",
```

```
        "utilization_rate": 0.87,
        "workload": "batch_inference_1k_tokens"
      },
      "teacher_model_ref": null
   },

  "privacy_security": {
   "pii_handling": "pseudonymized_at_source",
   "retention_days": 30,
   "security_controls": [
    "access_logging",
    "rate_limits",
    "model_abuse_monitoring"
   ],
   "privacy_risks": {
    "membership_inference_risk": "medium"
   }
  },

  "monitoring": {
   "post_market_metrics": [
    "drift_auc",
    "fairness_eod",
    "incident_rate"
   ],
   "alerting_sla_hours": 24,
   "fallback_mode": "disable_auto_rank_use_text_summary_only"
  },

  "conformity": {
   "documentation_ref": "doc:HS-TECHDOC-2.3.1",
   "audit_artifacts": [
    {
     "type": "internal_audit",
     "hash": "sha256:aa21...9d0c"
    },
    {
     "type": "third_party_report",
     "hash": "sha256:bb77...c18f"
    }
   ]
  },

  "attestations": {
   "signature": "sigstore:rekor:placeholder",
   "signing_key_id": "did:web:example.ai#compliance-key-1"
  }
}
```

*Listing 3.* Worked example of an AI Risk Manifest (YAML).

```
schema: nutrition-label.ai/manifest
schema_version: "1.0.0"
manifest_id: nutrition-label:demo:v1
issued_at: "2026-01-15T00:00:00Z"

issuer:
 org: ExampleAI Ltd.
 contact: compliance@example.ai

system:
 name: RawModel-1
 system_version: "2.3.1"
 model_type: LLM+ranker
 model_fingerprint: sha256:7c3b...e91a
```

```yaml
  deployment:
    mode: API
    regions:
      - EU
      - US
      - BD

intended_use:
  purpose: Decision support for recruiter triage of resumes
  users:
    - HR analysts
  decision_impact: employment
  human_in_the_loop: true
  prohibited_use:
    - fully-automated hiring decisions without human review
    - use outside declared job families without re-validation

risk_classification:
  eu_ai_act_style:
    tier: high_risk
    rationale: employment decision support
  nist_rmf_profile:
    functions:
      - GOVERN
      - MAP
      - MEASURE
      - MANAGE

data_provenance:
  training_sources:
    - type: public_job_postings
      region_scope:
        - EU
        - US
      license: mixed
    - type: historical_resumes
      region_scope:
        - EU
      license: internal_consent_basis
  sensitive_attributes_used: false
    known_gaps:
      - under-representation of candidates with non-traditional education
    security:
      data_filtering_pipeline: deduplication + PII scrub + toxicity filter v2.1
      poisoning_detection_methods:
        - activation_clustering
        - spectral_signatures
      dataset_integrity_checks: SHA-256 per shard, verified at load time

evaluation:
  eval_date: "2026-01-10"
  datasets:
    - name: HR-Triage-2025Q4
      region: EU
      n: 50000
      split: heldout
  metrics:
    utility:
      topk_precision@10: 0.61
      auc: 0.79
    robustness:
      ood_drop_auc: 0.06
      prompt_injection_pass_rate: 0.93

fairness:
```

```
 reporting_unit: EU legal categories where available
 group_metrics:
   - group: gender
     metric: equal_opportunity_diff
     value: 0.04
     threshold: 0.05
   - group: age_bucket
     metric: selection_rate_ratio_min
     value: 0.82
     threshold: 0.80
 mitigations:
   - post-hoc calibration
   - bias-aware re-ranking

energy:
 training_kwh: 42000
 inference_wh_per_1k_tokens: 2.1
 measurement_protocol: nutrition-label.ai/energy/v1
 measurement_context:
   hardware: NVIDIA A100-80GB
   utilization_rate: 0.87
   workload: batch_inference_1k_tokens
 teacher_model_ref: null

privacy_security:
 pii_handling: pseudonymized_at_source
 retention_days: 30
 security_controls:
   - access_logging
   - rate_limits
   - model_abuse_monitoring
 privacy_risks:
   membership_inference_risk: medium

monitoring:
 post_market_metrics:
   - drift_auc
   - fairness_eod
   - incident_rate
 alerting_sla_hours: 24
 fallback_mode: disable_auto_rank_use_text_summary_only

conformity:
 documentation_ref: doc:HS-TECHDOC-2.3.1
 audit_artifacts:
   - type: internal_audit
     hash: sha256:aa21...9d0c
   - type: third_party_report
     hash: sha256:bb77...c18f

attestations:
 signature: sigstore:rekor:placeholder
 signing_key_id: did:web:example.ai#compliance-key-1
```

The manifest (Table 3) standardizes *how* risk evidence is represented and exchanged, while jurisdictions retain authority over *what* thresholds and enforcement actions apply.

## B. Why Technical Schemas Can Succeed Where Political Treaties Fail

A recurring challenge in global AI governance is that binding political treaties are slow to negotiate, difficult to enforce, and often stall due to sovereignty concerns, geopolitical competition, or divergent economic priorities. By contrast, technical interoperability standards operate under a different incentive structure. Rather than requiring prior political alignment, they

*Table 3.* Crosswalk from AI Risk Manifest fields to (i) EU AI Act-style governance obligations (conceptual) and (ii) NIST AI RMF functions.

| Manifest Field | EU AI Act-style Obligation (Concept) | NIST AI RMF |
|---|---|---|
| `system.*` | System identification, versioning, traceability for technical documentation and auditability | GOVERN |
| `intended_use.*` | Defined intended purpose, user context, and prohibited uses (scope control; misuse prevention) | MAP |
| `risk_classification.*` | Risk tiering and regime applicability (high-risk triggers, conformity expectations, documentation depth) | GOVERN / MAP |
| `data_provenance.*` | Data governance: origin, licensing, representativeness gaps, and constraints relevant to bias and legality | MAP / MEASURE |
| `evaluation.*` | Evidence of performance and robustness testing under declared conditions (reproducible metrics) | MEASURE |
| `fairness.*` | Bias monitoring and non-discrimination reporting (disaggregated metrics + mitigation record) | MEASURE / MANAGE |
| `privacy_security.*` | Security/privacy controls, retention, and abuse monitoring consistent with risk controls and accountability | GOVERN / MANAGE |
| `monitoring.*` | Post-market monitoring: drift, incidents, corrective actions, and defined fallback behavior | MANAGE |
| `conformity.*`, `attestations.*` | Conformity evidence hooks: audit artifact hashes, third-party reports, and signed attestations | GOVERN / MANAGE |

diffuse through markets via adoption incentives, supply-chain pressure, and procurement requirements, often achieving de facto global reach without formal international agreements (Blind, 2016; UNCTAD, 2021).

This incentive-driven diffusion of technical standards is particularly salient in the current geopolitical context, where advanced AI systems are increasingly treated as strategic national assets rather than neutral commercial technologies. Recent work on Generative AI in the context of Industry 5.0 highlights how disparities in talent, compute, and data access are reshaping global power hierarchies and accelerating digital fragmentation (Wasi et al., 2025). As states weaponize export controls, data sovereignty, and industrial policy to secure AI advantage, formal treaty-based coordination becomes even less politically feasible. In this environment, interoperable technical schemas offer a rare coordination mechanism that aligns with sovereignty concerns while enabling cross-border accountability through market incentives rather than political compulsion.

In the AI context, firms, particularly those operating across borders, face strong incentives to adopt a single, reusable compliance artifact rather than maintain jurisdiction-specific documentation pipelines. For example, a U.S.-based company deploying AI products in Europe benefits directly from producing machine-readable risk manifests aligned with EU auditing and conformity expectations, as this reduces regulatory friction, accelerates market access, and lowers legal uncertainty. Once such artifacts are embedded into procurement requirements, certification regimes, or platform onboarding processes, adoption becomes economically rational even without legal compulsion (OECD, 2017). Importantly, this mechanism does not require U.S. regulators to enforce EU law; it relies instead on market access and transaction cost reduction as the primary drivers of convergence.

This pattern mirrors earlier successes in areas such as financial reporting standards, cybersecurity frameworks, and supply-chain transparency, where technical schemas spread through procurement, certification, and contractual norms rather than treaties (Funk & Methe, 2001). In this sense, interoperable AI risk manifests function as coordination infrastructure: they allow firms to comply once and reuse everywhere, making partial alignment economically preferable to fragmentation. The political feasibility of this approach lies precisely in its optionality: states retain legal autonomy, while firms voluntarily converge on shared technical representations because doing so is cheaper, faster, and more scalable than bespoke compliance (Krechmer, 2006).

## C. Verification, Attestation, and the Limits of Self-Reporting

A natural concern with any documentation-based governance mechanism is the risk of self-reporting bias, in which developers selectively disclose favorable metrics, omit unfavorable information, or conduct evaluations on unrepresentative test sets.

As noted in the schema, the manifest distinguishes *integrity*, guaranteed by cryptographic attestations, from *truthfulness*, which requires external validation. The mechanisms described here operationalize the third-party verification layer that bridges this gap. Without verification, AI risk manifests could degenerate into compliance theater, formally complete but substantively unreliable. This concern is well documented in prior work on AI accountability and documentation practices (Gebru et al., 2021; Raji et al., 2020). Our proposal addresses this limitation by treating manifests not as narrative reports, but as verifiable compliance objects.

Cryptographic attestations provide a concrete mechanism to mitigate misrepresentation. In practice, key fields in the manifest, such as model version identifiers, dataset hashes, evaluation metrics, and audit artifacts, are bound to cryptographic hashes and digitally signed by the producing entity or an accredited third party. While such signatures do not guarantee that the underlying evaluation is correct, they ensure immutability and accountability: once published, manifest contents cannot be altered without detection, and discrepancies can be traced to a specific actor and artifact (Spoczynski et al., 2025).

Crucially, this shifts the trust model. Regulators and downstream deployers are not required to trust the developer's claims directly; they only need to trust the integrity of the verification chain. Independent auditors or certified testing bodies can issue attestations referencing the same hashed artifacts, enabling cross-checking without re-running full evaluations. Although this does not eliminate garbage-in risks entirely, it substantially raises the cost of deception and enables post hoc accountability by making false claims provable rather than merely disputable (Crilly, 2025).

In this design, verification is incremental rather than absolute. The manifest establishes a minimum verifiable baseline upon which stronger guarantees, such as reproducible evaluation environments, third-party audits, or regulatory spot checks, can be layered. This mirrors established practices in software supply-chain security and financial reporting, where cryptographic integrity and auditability do not prevent all misconduct but significantly reduce its feasibility and impact (Manski, 2022).

## D. Addressing the Innovation vs. Standards Dilemma

### D.1. Counterargument: Standards Will Lag Behind Innovation

A common concern voiced by critics is that regulations, including standards, will *stifle innovation and progress* by slowing down AI advancements and creating barriers to entry for new companies, potentially strengthening incumbents (Cohen, 2019). These critics argue that rigid regulations may hinder AI's inherent adaptability and its ability to learn from new data, thereby slowing the development and deployment of beneficial AI applications. The rapid pace of AI development, described as *accelerated* and making it *even harder for the already trailing legal system to catch up*, fuels this argument (Harvard Law Today, 2025). There is a fear that if one country imposes stringent regulations while others adopt a more flexible approach, it could disadvantage domestic companies in the global AI race (Heller, 2023). The decentralized nature of AI innovation, often driven by a multitude of individual contributors and for-profit enterprises, further complicates traditional, often slower, regulatory mechanisms (Ananny & Crawford, 2018).

### D.2. Response: Modular, Versioned Standards for Agile Evolution

In our view, the notion that standards inevitably lag behind innovation fails to account for the emergence of agile, modular approaches to standardization, an evolution we believe is not only possible but necessary. While we acknowledge that regulation can indeed impede progress if poorly designed or overly prescriptive, we contend that the absence of common, interoperable standards in a globalized AI market can pose even greater barriers to progress. Rather than promoting agility, regulatory fragmentation often results in redundant engineering, inconsistent compliance burdens, and constrained market reach, especially for emerging players. If each jurisdiction requires different reporting formats and risk categories, developers must adapt the same AI system multiple times, slowing innovation and limiting scalability. In this sense, the very absence of standards becomes the bottleneck.

To counter this, we advocate for a standards framework that is flexible, inclusive, and capable of evolving in parallel with the technological landscape. We propose several key strategies to realize this vision:

**Modularization:** We believe that modular design is key to ensuring that standards remain adaptable. By decomposing complex systems into interoperable components, each governed by its own evolving specification, we enable isolated updates that avoid triggering systemic overhauls. For instance, standards for model bias assessment can evolve independently from those governing data provenance or energy usage. This enables targeted innovation and iterative improvement without forcing costly re-certification of the entire system. In our experience, this design philosophy closely mirrors the success seen

in software engineering, where decoupled modules allow for rapid iteration and experimentation (Parnas, 1972).

**Versioning and Iterative Development:** Just as we update software systems to keep pace with user needs and security vulnerabilities, we argue that AI standards should adopt a versioned, iterative model (Beck et al., 2001). Through regular updates and feedback loops, standards can remain current and responsive to emerging challenges and opportunities (van Belkom et al., 2020). Drawing inspiration from the Agile Manifesto, we suggest that this process emphasizes collaboration, responsiveness to change, and continuous refinement. In our work, we have seen that this agile approach significantly reduces the cost of late-stage corrections and fosters a culture of continuous improvement, which is essential for AI systems deployed in high-stakes environments.

**Community and Open Source:** We emphasize the value of open standards developed through collaborative, transparent processes (OpenStand Principles, 2012). In our view, community-driven ecosystems, like those that underpin successful open-source software, can dramatically enhance the quality, usability, and security of standards. By inviting public scrutiny and contributions, we can identify blind spots, address diverse stakeholder needs, and accelerate adoption. We also advocate for the use of open-source software tools to implement and validate these standards, ensuring accountability and trust across different actors, including governments, civil society, and industry (Whittaker et al., 2018).

**Focus on Outcomes, Not Prescriptive Methods:** Rather than dictating specific technical implementations, we argue that standards should be centered around clearly defined outcomes. For example, it is more constructive to require that an AI system meet a threshold for fairness or energy efficiency than to mandate a specific model architecture or mitigation technique. This outcome-oriented approach allows developers the creative freedom to pursue novel solutions within a well-defined ethical and safety perimeter. We believe this strikes the right balance between enabling innovation and ensuring responsible deployment.

In summary, we urge that standards be seen not as constraints but as enabling infrastructure, akin to standardized network protocols or financial APIs that have historically unlocked transformative innovation. Globally harmonized AI risk standards can serve a similar role by offering a common *tech stack* for safety, transparency, and compliance. This infrastructure reduces redundant efforts, simplifies cross-border deployment, and levels the playing field for innovators regardless of size. In our view, reframing standards as tools for scalability and trust, not merely compliance, will be key to accelerating inclusive, responsible AI development worldwide.

## E. Overcoming the “Cold Start” Problem

**Timeline and convergence expectations.** Full global consensus is neither required nor expected for the proposed framework to function; partial convergence around a minimal interoperable schema is sufficient for procurement, auditing, and compliance workflows across jurisdictions. We anticipate the following staged progression. Within 1-2 years, major public-sector procurers (e.g., US DoD, EU agencies) begin mandating machine-readable risk disclosures as contract prerequisites. Within approximately 3 years, major cloud providers converge on a unified schema to reduce cross-border compliance overhead, establishing a de facto industry standard. Within 4-5 years, formal standards bodies such as ISO/IEC may ratify the schema, though this is an upper-bound scenario rather than a necessary condition. Historical analogies support this timeline: GDPR moved from proposal (2012) to enforceability (2018) over six years despite strong political backing; ISO 27001 achieved broad enterprise adoption over roughly a decade following its 2005 formalization. Governance standardization is constrained by procurement cycles and institutional ratification, not model development speed.

### E.1. A Six-Stage Cold Start Pathway

To make this diffusion concrete, we propose the following sequenced adoption strategy that bypasses the need for prior international treaty negotiation.

*(1) Public-sector procurement as catalyst.* Major governmental bodies (e.g., US Department of Defense, European Commission agencies) embed the machine-readable manifest as a mandatory prerequisite for AI software acquisition. Concentrated demand from high-value contracts incentivizes foundation model developers to produce manifests to maintain market access.

*(2) Cloud provider integration.* Major cloud providers (AWS, Azure, Google Cloud) integrate the schema into model registries and API gateways to support compliance and liability workflows for their enterprise customers, embedding the schema throughout the supply chain.

*(3) Open-source tooling commoditization.* The open-source community builds automated generation, parsing, and validation

tooling, reducing the compliance cost to near zero and removing the resource barrier for SMEs and academic researchers.
*(4) Sovereign and regional convergence.* Smaller nations and regional blocs (e.g., ASEAN) adopt the de facto standard rather than bear the cost of designing proprietary regulatory technology, enforcing their specific legal thresholds through the shared technical substrate.
*(5) Institutional formalization.* ISO/IEC ratifies the schema following de facto convergence, providing long-term multi-stakeholder stewardship without requiring a preliminary treaty.
*(6) Iterative extension.* Domain-specific modules (healthcare, finance, agentic systems) are added through governed extension blocks, preserving core interoperability primitives while accommodating specialized regulatory requirements. This pathway does not require geopolitical consensus as a precondition; it uses market access and transaction cost reduction as the primary convergence drivers.