# OpenReview forum: "Position: AI Governance Needs ISO-like Interoperability Protocols, Not Just Laws"
_ICML.cc/2026/Position_Paper_Track — ICML 2026 Position Paper Track spotlight_

### Official Review · Reviewer_VCB4 · 2026-02-25

**Significance:** 3
**Argument Clarity:** 3
**Ethics Flag:** Yes
**Rating:** 4
**Confidence:** 4

**Questions:**

See the weaknesses above. Basically, I agree with the authors that AI governance is important, but I have doubt on the probability to achieve the ultimate consensus. I am wondering if the authors have a timeline prediction?

**Alternative Views Section:**

Yes

**Compliance With Llm Reviewing Policy A Conservative:**

Affirmed.

**Discussion Potential:**

3

**Ethical Review Concerns:**

The studied topic is closely related to EU AI Act, GDPR, etc.

**Ethics Review Area:**

["Legal Compliance (e.g., EU AI Act, GDPR, copyright, terms of use)"]

**Final Justification:**

I think the authors have addressed an important problem at the era of AI, but I have some concerns about the feasibility to form a protocol.

**Paper Summary:**

This paper argues that AI governance must be built not on laws alone, but on ISO-like interoperability protocols that enable standardized,
machine-readable risk communication across borders. It proposes the term "AI nutrition labels" containing unified metrics for bias, energy
usage, and data provenance to facilitate cross-jurisdictional compliance.

**Position:**

Yes

**Position In Title:**

Yes

**Related Work:**

2

**Strengths And Weaknesses:**

Strengths:
1. I agree with the authors that AI governance is an important problem and needs interoperability protocols.
2. The AI nutrition labels are designed based on a minimal, machine-readable principle.
3. The discussion and alternative views are reasonable.

Weaknesses:
1. As pointed out, AI systems increasingly update through continuous learning and deployment, critics contend that any fixed standard risks becoming obsolete before it is widely adopted. This seems difficult to solve.
2. Why would big companies like to uncover their detailed "AI nutrition labels"? For example, none of current proprietary LLMs reports their training data.
3. Maybe SBOM (Software Bill of Materials) can be referenced in addition to GDPR.

**Support:**

3

---

> ### Author Rebuttal · Authors · 2026-03-30
>
> We sincerely thank the reviewer for the positive evaluation and for recognizing the importance of interoperable AI governance protocols and the clarity of our proposed machine-readable AI nutrition labels. We appreciate the acknowledgment of the minimal design principle and the balanced discussion of alternative views.
>
> ---
>
> Below, we address each of your concerns and questions in detail:
>
> - **W1: Continuous Learning and Obsolescence Risk.** Thank you for highlighting this critical challenge. We agree that overly rigid, fixed standards risk rapid obsolescence, which we acknowledge in Section 6 (Alternative View 1). However, avoiding standards altogether leads to unmanageable fragmentation. This is precisely why ongoing, collaborative discussions are necessary to develop a standard that will endure. To solve this issue, our aim is to build the schema on "modular, versioned, and outcome-oriented" primitives (Section 6 and Appendix D.2) rather than prescriptive technical methods. By decoupling specific modules (e.g., updating bias metrics without breaking energy reporting), the standard can iteratively adapt to continuous learning systems.
>
> - **W2: Proprietary Concerns and Big Companies.** Thank you for this practical question. It is true that many proprietary LLM developers guard their training data closely. However, our proposed manifest does not ask companies to reveal exact datasets, model weights, or algorithmic trade secrets. Instead, as outlined in Section 3.2, the `data_provenance` metric only requires a summary describing data sources, geographic scope, and applicable licensing. While some high-level metadata (like broad data categories or model size) will be disclosed (such as number of parameters), this level of transparency is standard for safety and governance compliance across all regulated industries. We expect big companies to provide these nutrition labels because major buyers, especially public-sector procurement, will require them to secure lucrative contracts, just like how the ISO standardization and balancing works, along with legal requirements (Section 4).
>
> - **W3: Referencing SBOM (Software Bill of Materials).** Thank you for this excellent suggestion. We actually briefly mention Software Bills of Materials in Section 3.1 (comparing our manifests to them) and Section 4 (noting their role in IoT governance). However, we completely agree that the SBOM framework deserves a much more prominent, detailed comparison alongside our GDPR discussion. SBOMs are the perfect real-world analogy for how interoperable, machine-readable artifacts become an industry standard through supply-chain pressure rather than top-down laws.
>
> * **Q1. Timeline Prediction for Consensus.** Thank you for this forward-looking question.
>   - While binding international treaties may take decades, we expect faster technical convergence driven by market forces rather than formal political agreement. Based on the Cold Start strategy, we anticipate a **1–2 year horizon** for major public-sector procurers (e.g., US DoD, EU agencies) to begin mandating basic machine-readable risk disclosures. Within **~3 years**, major cloud providers are likely to converge on a unified schema to reduce cross-border compliance overhead, establishing a de facto industry standard. Within **4–5 years**, formal bodies such as ISO/IEC may ratify this schema, though we frame this as an upper-bound scenario rather than a required outcome. Importantly, full global consensus is not required; partial convergence around a minimal interoperable schema is sufficient for procurement, auditing, and compliance workflows across jurisdictions.
>   - While this may appear slow relative to AI capability progress, governance standardization is constrained by procurement cycles, institutional adoption, and formal ratification rather than model development speed. The estimate reflects sequential stages: procurement adoption (1–2 years), platform alignment (~3 years), and standards-body formalization, which typically follows de facto convergence. Historical examples support this pattern: GDPR (proposal 2012, adopted 2016, enforceable 2018) required several years from proposal to compliance despite strong political backing, and ISO 27001 (formalized 2005, broad adoption 2005–2015) similarly evolved through iterative enterprise uptake before becoming a baseline standard.
>   - We will add a clarifying paragraph in Section 5 emphasizing that partial convergence, rather than perfect global consensus, is sufficient for the proposed framework to function.
>
>
> ---
>
> We will incorporate these clarifications in the final version of the paper. We believe they further strengthen the scope, feasibility, and technical grounding of the proposed framework.

---

> > ### Author Rebuttal · Reviewer_VCB4 · 2026-04-03
> >
> > See my comment above.

---

### Official Review · Reviewer_Z5wY · 2026-03-08

**Significance:** 3
**Argument Clarity:** 3
**Rating:** 5
**Confidence:** 2

**Questions:**

Could the authors discuss possible extensions of the nutrition labels?

**Alternative Views Section:**

Yes

**Compliance With Llm Reviewing Policy A Conservative:**

Affirmed.

**Discussion Potential:**

3

**Paper Summary:**

The paper calls for machine-readable ISO-like "AI nutrition label" file, which reports AI compliance, risks and impacts. After reviewing the case of  GDPR–ISO 27001, it provides an illustrative example of such a file (in json), with entries that include fairness measures, environmental impacts and data provenance. The paper then argues that the worldwide standardization of such files would allow to both facilitate AI regulation law enforcement and developers' AI compliance efforts.

**Position:**

Yes

**Position In Title:**

Yes

**Related Work:**

3

**Strengths And Weaknesses:**

Strength:
- The proposal is clear and well supported.

Weakness:
- I would have appreciated more thoughts on what the file standard could initially be, and how it could potentially evolve in the future. Could there also be entries on past legal cases, emotional attachment, sycophancy, privacy measures used when training, security measures against poisoning, security measures against prompt injections, and so on?
- Perhaps a related call to be added is on research on the evaluations that must/should be reported in the AI nutrition labels?

**Support:**

3

---

> ### Author Rebuttal · Authors · 2026-03-30
>
> We sincerely thank the reviewer for the positive evaluation and for recognizing the clarity and strong support of our proposal for machine-readable ISO-like “AI nutrition labels” to enable standardized reporting of AI compliance, risks, and impacts.
>
> ---
>
> Below, we address each of your concerns and questions in detail:
> - **W1: Initial Standard, Evolution, and Specific Entries.** Thank you for these constructive suggestions.
>    - The paper already outlines an initial file standard in Section 3.4, with a baseline schema covering model identity, intended use, data provenance, fairness, and energy metrics. The schema is designed to be modular and versioned, specifying what must be demonstrated rather than how it must be implemented, which supports gradual evolution as new requirements emerge.
>    - Regarding specific entries, the current YAML/JSON schema (Appendix A) already includes fields for privacy and security (e.g., `pii_handling`, `security_controls` such as `model_abuse_monitoring`) and robustness to prompt injection (`prompt_injection_pass_rate` in the `evaluation` block). To address data poisoning, we agree this should be made more explicit. The schema can incorporate fields such as `data_filtering_pipeline`, `poisoning_detection_methods`, and `dataset_integrity_checks`, along with corresponding evaluation metrics (e.g., adversarial contamination tests or robustness scores). These would document both preventive controls and empirical validation.
>    - We also agree that entries related to legal precedent, emotional attachment, or sycophancy are valuable extensions. These are well suited as future modular additions rather than core baseline requirements. We will clarify both the current coverage and these potential extensions, including poisoning defenses, in the final version.
> - **W2: Call for Research on Evaluations.** Thank you for this excellent recommendation. Currently, our Call to Action (Section 7) includes three recommendations: establishing a shared technical baseline (R1), tying standards to incentives (R2), and investing in multi-stakeholder standard stewardship (R3). While R1 implicitly touches on defining what goes into the baseline, we completely agree that it lacks an explicit call for the *research community* to develop the underlying evaluation science. To address this, we will add a new recommendation, **"R4: Fund and formalize the science of AI evaluation,"** in Section 7 of the final manuscript, explicitly calling on the research community to prioritize developing robust, standardized evaluation methodologies for emerging risks, ensuring that the metrics populating these manifests are scientifically rigorous and reproducible.
>   - *R4: Fund and formalize the science of AI evaluation.* The utility of an interoperable AI Risk Manifest depends on the scientific validity and reproducibility of the metrics it encodes. Unlike traditional software, AI systems are opaque, adaptive, and probabilistic, making static evaluation difficult. Current methods for assessing emergent behaviors such as prompt injection, socio-technical bias, and agentic drift remain fragmented and underdeveloped. Policymakers, funding agencies, and industry consortia should invest in the foundational science of AI evaluation, including robust mathematical benchmarks for fairness, hardware-agnostic protocols for energy measurement, and rigorous automated red-teaming methods. Without scientifically grounded and hard-to-game metrics, the manifest risks enabling compliance theater rather than meaningful accountability. Therefore, schema standardization must be coupled with sustained investment in the evaluation science that informs it.
>
> - **Q1: Possible Extensions of the Nutrition Labels.** Thank you for this forward-looking question. The JSON/YAML manifest is fundamentally designed as an extensible container, allowing for optional extension blocks while preserving stable interoperability primitives. Beyond the core metrics detailed in Section 3.2, the labels could be extended through domain-specific or capability-specific modules. For instance, a healthcare extension could include clinical validation cohorts, demographic distribution of trial data, and medical device clearance status. A financial extension could include stress-testing results against macroeconomic distribution shifts. Furthermore, as AI agents become more autonomous, *agentic extensions* could capture operational constraints, detailing allowable action spaces, API permissions, and mechanisms that *enable systems to decline unsafe actions or exit hazardous operational states*.
>
> ---
>
> Thank you again. We will incorporate these clarifications in the final version of the paper. We believe they further strengthen the scope, feasibility, and technical grounding of the proposed framework.

---

> > ### Author Rebuttal · Reviewer_Z5wY · 2026-04-05
> >
> > Thank you for the rebuttal. I have no further question.

---

### Official Review · Reviewer_CPLX · 2026-03-12

**Significance:** 2
**Argument Clarity:** 4
**Rating:** 5
**Confidence:** 4

**Questions:**

1. Can you clarify how this protocol is verifiable? If a lab reports "energy_use": { "training_co2_tons": 0}. how would this be verified?
2. How do you propose a distilled model should be reported? Should it include the energy cost of the teacher model?
3. Does the training energy include all checkpoints and variations or only the "final" training run?
4. Why would an organisation report this data for their model if they are solely operational in a jurisdiction with lax regulations? Or include fields that are not mandatory for their region?
5. How would you balance updating the protocol to keep it fit for purpose vs undermining the standardisation it is meant to bring?

**Alternative Views Section:**

Yes

**Compliance With Llm Reviewing Policy A Conservative:**

Affirmed.

**Discussion Potential:**

4

**Final Justification:**

The authors addressed my concerns. I believe the clarifications the authors will adopt strengthen the paper.

I will keep my rating of **Accept**

**Paper Summary:**

The paper argues that for effective AI governance, laws are insufficient and that there is a need for ISO-style protocols to create a standardised AI *nutritional labels* for effective adherence. Their main arguments include the need for computer readable protocols, simplicity, adaptability and standardisation. They highlight the success of GDPR and its corresponding ISO standard (ISO 27001) which the authors argue was one of the major reasons for its success.

**Position:**

Yes

**Position In Title:**

Yes

**Related Work:**

3

**Strengths And Weaknesses:**

## Strengths

- The paper is well written and their position is clear.
- The idea of a simple standardised cross-jurisdictional protocol or so called AI *nutritional labels* is well motivated and would likely provoke discussion in the community.
- They include an example using their proposed protocol showcasing how a model is reported. Well thought-out versioning.
- The authors propose to include a signed hash to the report allowing it to bind the specific model to the reported values which would prevent tampering and allow for audit trails.
- They present two fair counter arguments but questions still remain which could allow for future discussion.

---

## Weaknesses

- Some elements in their proposed protocol seem too simple to capture some novel work. For example, their protocol requires reporting inference-time energy consumption under a standardised scheme. This scheme specifies things such as hardware for easy comparison between systems. However, this approach would likely bias some model architectures that are designed for the tested hardware vs other more efficient algorithms designed for different hardware. This is a similar point to their Alternative View 1, but I find their followup reasoning lacking.
- It is unclear how the proposed protocol captures distilled models. Both regarding data and energy usage.
- The authors reasoning for why this protocol will not simply end up being the n+1 competing standard is on the weaker side. There are proposed protocols already (as mentioned in the paper) which have not become the de facto. The authors explanation for this is the lack of computer-readable protocol and fragmentation. However, many of the successful ISO protocols are not naturally computer readable. And as for their example with GDPR, ISO 27001 is not sufficient and organisations often add ISO 27701 plus additional evidence.
- As the authors state, different jurisdictions have varying restrictions and definitions for compliance. With some strict and others voluntary. In jurisdictions with limited restrictions, expecting people to voluntarily follow this protocol is a big ask. Decreasing the likelihood of this protocol becoming a universal success.
- The authors mention the protocol needs to adapt as the field progresses but no discussion is included on the negative aspects of ever-changing protocols. Some acknowledgement of the compromises regarding protocols that update often would strengthen the paper.
- Minor point: Some sections feel repetitive. (lines 28-45 right) (lines 179-185 right) (lines 284-300 right)

**Support:**

3

---

> ### Author Rebuttal · Authors · 2026-03-30
>
> We sincerely thank the reviewer for the positive evaluation and for recognizing the clarity and motivation of our work on AI governance protocols that follow ISO standards and machine-readable nutrition labels. We appreciate the acknowledgment of our versioned reporting structure and signed hash mechanism for auditability and tamper resistance, as well as the recognition of cross-jurisdictional standardization and constructive counterarguments.
>
> ---
>
> * **W1: Hardware Bias in Energy Reporting.** Thank you for this insightful comment. We agree that transparency alone does not resolve comparability concerns, as standardized measurement can still advantage certain architectures or deployment setups. Our approach is to improve comparability by requiring disclosure of the full measurement context, including hardware type, utilization rates, workload configuration, and measurement methodology, rather than reporting a single aggregate value. This allows evaluators to normalize or reinterpret energy metrics across systems. The goal is not to enforce a single “fair” metric, but to enable consistent, context-aware comparison while avoiding bias introduced by incomplete reporting.
> - **W2/Q2: Capturing Distilled Models.** Thank you for raising this excellent point regarding distilled models. Yes, a distilled model's manifest should absolutely reflect its lineage. We propose extending the `data_provenance` and `energy_use` schema blocks to include a `teacher_model_ref` field. This would allow the distilled model to inherit or explicitly link to the training energy cost and data provenance of its teacher, while reporting its own (significantly lower) inference costs separately. This maintains the traceability of the entire AI supply chain.
> - **W3: Risk of Becoming the "n+1" Standard.** Thank you for this critique. We agree that successful ecosystems typically consist of layered standards rather than a single dominant framework. Our intent is not to replace existing approaches, but to define a minimal interoperability layer that can coexist with and link to companion standards, domain-specific schemas, or regulatory annexes. The novelty lies in combining machine-readability (e.g., JSON/YAML) with an explicit regulatory crosswalk, enabling integration into MLOps pipelines and procurement systems. In this sense, the manifest functions as a unifying substrate that reduces coordination costs while remaining compatible with a broader standards stack.
> - **W4/Q4: Voluntary Adoption in Lax Jurisdictions.** Thank you for this pragmatic concern. While we cannot enforce adoption in jurisdictions without binding rules, uptake is driven by market and supply chain incentives, similar to ISO standards. Organizations seeking access to multinational enterprises, major cloud providers, or public procurement (e.g., DoD or EU agencies) will be required to provide the standardized manifest. Thus, the standard achieves global reach through market access gates, even where local laws are weak.
> - **W5/Q5: Balancing Protocol Evolution and Stability.** Thank you for highlighting this crucial tension. Updating protocols too frequently can indeed undermine the standardization it seeks to create, causing compliance fatigue. We propose managing this tension through an international governing body (such as a consortium involving ISO, IEC, and IEEE) that issues major schema versions (e.g., v1.0, v2.0) on a measured, multi-year cycle, while allowing localized, modular extensions for rapid field-specific changes. This mirrors how traditional ISO standards evolve, maintaining stable core primitives while updating specific annexes as technology progresses.
> - **W6: Repetitive Sections.** Thank you for noting this. We will streamline the paper to improve flow and accommodate the above additions.
> - **Q1: Verifiability of Claims.** Thank you for this question. The manifest does not inherently detect false reporting. Instead, following ISO practice, it relies on independent third-party evaluation. As in Appendix C, cryptographic attestations (signed hashes) bind reported metrics to certified audit bodies or external testing reports. Verifiers do not trust raw claims but validate auditor signatures attached to each claim.
> - **Q3: Scope of Training Energy Reporting.** Thank you for asking for this clarification. We propose that the reporting should be as detailed as possible, capturing the entire training lifecycle. However, to ensure feasibility, the schema defines a minimum baseline, mandating the reporting of the energy cost for the *final* training run of the deployed model checkpoint. The schema will also include optional, structured fields to report the cumulative energy of all experimental checkpoints, hyperparameter tuning, and variations.
>
> ---
>
> We will incorporate these clarifications in the final version of the paper. We believe they further strengthen the scope, feasibility, and technical grounding of the proposed framework.

---

> > ### Author Rebuttal · Reviewer_CPLX · 2026-04-03
> >
> > The authors addressed my concerns. I believe the clarifications the authors will adopt strengthen the paper.
> >
> > I will keep my rating of **Accept**

---

### Official Review · Reviewer_H3tF · 2026-03-13

**Significance:** 3
**Argument Clarity:** 4
**Rating:** 4
**Confidence:** 4

**Questions:**

Q1. Traditional IT standards like ISO 27001 apply to deterministic systems. How can a static "AI Nutrition Label" effectively capture the stochastic and emergent risks of AI models that may only surface during runtime or under distributional shifts not covered by the initial manifest certification?

Q2. How does the framework mitigate the risk of **Goodhart’s Law**, where developers might optimize their models specifically to "pass" the manifest’s standardized keys rather than genuinely reducing risk? Is there a provision for "dynamic" or "probabilistic" auditing to counter adversarial reporting?

Q3. In the absence of a global enforcement body, what would be the proposed "Cold Start" strategy? Why would dominant AI actors or sovereign nations with competing geopolitical interests voluntarily converge on this standard rather than creating their own proprietary or regional alternatives?

Q4. While the paper suggests that standardized manifests can accommodate different legal thresholds, it remains unclear how the schema handles mutually exclusive mathematical definitions of core concepts like fairness. Specifically, since achieving compliance with one jurisdiction's metric may mathematically precludes compliance with another's, how does a unified JSON substrate prevent semantic fragmentation, where the same 'key' carries fundamentally different meanings across borders? Does the framework include a 'Meta-Ontology' to map these conflicting definitions, or is it merely a container for unharmonized data?

**Alternative Views Section:**

Yes

**Compliance With Llm Reviewing Policy A Conservative:**

Affirmed.

**Discussion Potential:**

4

**Final Justification:**

The rebuttal addressed my questions, and I would like to raise my score to 4.

**Paper Summary:**

The paper argues that as AI scales globally, law alone can't keep up with its risks. To fix this, the authors suggest a technical interoperability layer that turns vague ethics into automated, machine-readable checks. Their main idea is a metadata schema (an AI Nutrition Label) that standardizes how we report on things like bias and data sources. The framework maps technical data directly to legal rules. These protocols are modular and versioned, allowing them to evolve alongside new technology. This setup enables auditing across different borders, and lets governance catch up with the pace of the AI itself.

**Position:**

Yes

**Position In Title:**

Yes

**Related Work:**

3

**Strengths And Weaknesses:**

Strengths
1. The paper addresses the transition from high-level ethical principles to verifiable MLOps practices, which is an important and timely bottleneck in the AI community.

2. The comparison between GDPR (law) and ISO 27001 (implementation) provides a historical precedent for the proposed framework.

3. Unlike many position papers that remain purely philosophical, this paper proposes the AI Risk Manifest, and demonstrates a clear mapping between technical metadata and high-level legal requirements (e.g., mapping data hashes to GDPR/AI Act provenance requirements).

4. The emphasis on "machine-readability" is a major strength. It acknowledges that at AI scale, human-led auditing is a physical impossibility, and automated verification provides possibility for governance to keep pace with AI innovation.

Weaknesses
1. The paper lacks a detailed discussion on Verifiable Computing or mandatory third-party "ground truth" testing. Although in Appendix C, the paper discusses how cryptographic hashes ensures Integrity, it is less clear on ensuring truthfulness. A manifest can be internally consistent and untampered, yet still report 'false' metrics if the underlying evaluation was performed on a cherry-picked or biased test set. The paper lacks a rigorous discussion on how to bridge this 'truthfulness gap'.

2. The comparison to ISO 27001 is helpful but slightly flawed. ISO 27001 governs organizational processes (how a company behaves), whereas the AI Manifest governs product outputs (how a model behaves). The paper glosses over the fact that AI behavior is stochastic and emergent, making it much harder to "standardize" than traditional IT security processes.

3. There lacks a "Cold Start" strategy. In a world of geopolitical tension, it is unclear why dominant AI players or sovereign nations would agree on a single metadata standard without an international enforcement body.

**Support:**

2

---

> ### Author Rebuttal · Authors · 2026-03-30
>
> We sincerely thank the reviewer for recognizing the relevance of our work on bridging ethical principles and verifiable MLOps practices. We appreciate the acknowledgment of the proposed AI Risk Manifest and the importance of machine-readable governance mechanisms for scalable auditing and cross-border interoperability and linking technical metadata with legal requirements.
>
> ---
>
> Below, we address each of your concerns and questions in detail:
>
> * **W1: Integrity vs. Truthfulness (Verifiable Computing).** Thank you for this important distinction. Cryptographic integrity does not guarantee truthfulness, and the manifest is intended as a minimum verifiable baseline rather than a truth oracle. The current design already incorporates this through `data_provenance`, `evaluation`, and cryptographic attestation fields, as well as the layered verification logic described in Appendix C. These elements enable third-party audits and external validation of reported claims. We will connect Apx. C and clarify third-party verification.
> - **W2: Limits of ISO 27001 Comparison.** Thank you for this insightful critique. We completely agree that AI behavior is stochastic and emergent, which is why we explicitly state in Section 2 that the GDPR-ISO 27001 model cannot be transferred wholesale to AI governance because AI systems are probabilistic. Our analogy to ISO 27001 is meant to illustrate the *mechanism* of translating legal text into a shared technical operational language, rather than perfectly equating IT processes with AI product outputs. To handle emergent AI behaviors, our manifest goes beyond static process checks and incorporates continuous `monitoring` fields, including `post_market_metrics` (like drift and incident rates) and specific `fallback_mode` behaviors.
> - **W3/Q3: Geopolitics and Cold Start Strategy.** Thank you for raising this practical concern. While geopolitical tensions make binding international treaties difficult, we have not ruled out the role of an international body; in fact, we explicitly advocate for anchoring this effort in existing institutions like ISO and IEC. We argue that sovereign nations and dominant actors will converge on this not out of political goodwill, but because of market incentives and the sheer cost of fragmented compliance. Multinational firms will push for a unified schema to reduce the friction of deploying the same model across the US, EU, and Asia. therefore, our cold start strategy bypasses global political consensus by leveraging buyer power through public-sector procurement. First, major government bodies (e.g., US DoD and EU agencies) make the machine-readable manifest mandatory for software acquisition. Second, cloud providers integrate the schema into model registries and API gateways to support compliance and liability workflows. Third, this demand incentivizes foundation model developers to generate manifests to access high-value markets. Fourth, open-source communities build tooling that reduces generation cost to near zero. Fifth, once established as a de facto standard, smaller nations adopt it due to existing infrastructure and low cost. Finally, ISO/IEC formalizes the standard without requiring prior treaty negotiation. We will expand this pathway in the final paper due to rebuttal space limits.
> - **Q1: Static Labels vs. Stochastic Risks.** Thank you for the question. The manifest is a versioned, evolving artifact (`schema_version`, `system_version`) rather than a static label. It captures runtime risks via mandatory `monitoring` protocols and `post_market_metrics` (e.g., `drift_auc`, `incident_rate`) plus `fallback_mode` definitions. It standardizes reporting of ongoing risk management, not fixed snapshots.
> - **Q2: Goodhart’s Law and Adversarial Reporting.** Thank you for this point. The manifest avoids metric fixation by requiring disclosure of selected metrics and rationale, rather than enforcing a single scoring rule. Like ISO audits, compliance is reinforced through continuous review and certificate revocation mechanisms. We further propose dynamic auditing via API-accessible telemetry for probabilistic third-party verification, mitigating adversarial reporting.
> - **Q4: Semantic Fragmentation and Mutually Exclusive Metrics.** Thank you for this question. The manifest does not attempt to resolve conflicting legal or mathematical definitions of fairness. Instead, it standardizes the *representation* of metrics by requiring explicit reporting of the exact definition used (e.g., `equal_opportunity_diff`), subgroup specification, and raw values. This ensures that the same JSON artifact can be interpreted differently by EU and US regulators according to their respective legal thresholds. In this sense, the schema functions as a neutral, machine-readable reporting layer rather than a harmonizing ontology.
>
> ---
>
> We believe these clarifications strengthen the scope, feasibility, and technical grounding of the proposed framework.

---

> > ### Author Rebuttal · Reviewer_H3tF · 2026-04-05
> >
> > Thanks for the detailed response. My questions have been solved. And I will raise my score.

---

### Decision · Program_Chairs · 2026-04-30

**Decision:**

Accept (spotlight)

**Comment:**

All reviewers agree that this is an important topic for the community and the paper is well written, especially with improvements suggested during the review process.